# Collembola from Two Samplings in the MSS of the Sierra de Guadarrama National Park in Two Different Seasons, with Description of a New Species [note 1]

**DOI:** 10.3390/insects13111025

**Published:** 2022-11-06

**Authors:** Enrique Baquero, Rafael Jordana

**Affiliations:** 1Department of Environmental Biology, Faculty of Sciences, University of Navarra, University Campus, 31080 Pamplona, Spain; 2Institute for Biodiversity and Environment BIOMA, University of Navarra, Irunlarrea 1, 31008 Pamplona, Spain

**Keywords:** springtails, new species, mesovoid shallow substratum (MSS), subterranean sampling devices (SSDs), Iberian Peninsula

## Abstract

**Simple Summary:**

The interest in the fauna of the colluvial mesovoid shallow substratum (MSS) led us to install a series of pitfall traps (subterranean sampling devices or SSDs) for a full year in the Sierra de Guadarrama. This paper presents the comparative results between the captures at two different periods of the year and allows the description of a new species, found only in one of those periods. It seems proven that there are species present throughout the year, but others also predominate, or are exclusive, to just one of them. This study indicates, again, that the colluvial MSS has a particular species composition for the taxon Collembola, different from the surface, a perfectly differentiated habitat.

**Abstract:**

An intensive sampling in a colluvial mesovoid shallow substratum (MSS) of the Sierra de Guadarrama National Park, using 33 subterranean sampling devices (SSDs) is the origin of the Collembola studied in this paper. The data were obtained from the second extraction of the traps, in operation between October of 2015 and May of 2016. This paper presents the faunistic and diversity data along with the entire park (mostly at sampling points above 200 m a.s.l.) for this period, compares the data between the first extraction of the traps and the second one, and describes one species of the genus *Pseudosinella* that appears as new in the second campaign.

## 1. Introduction

The mesovoid shallow substratum (MSS) consists of a network of interstices and subsoil fissures, and harbors diverse epigean species of a stenoic nature, and strictly hypogean species, permanent inhabitants of this environment [1,2]. Previous studies were focused on ecology [3,4], or faunal aspects [5,6,7]. Some studies on the MSS have been carried out in karstic areas. In a karstic environment, there are caves, and the scree-slope can be an intermediate place between the caves and the outside environment. The springtails that live in caves are not really from caves but from the infractions of the karstic terrain, which by definition is very broken. There are no caves in terrains such as the MSS de Guadarrama. In this paper, a continuation of five previously published [8,9,10,11], the fauna found in the same traps but in different periods of the year is compared, for a complete season. In addition to an important substitution of species between seasons, in the second period, a new species for science (*Pseudosinella*) that had not been captured in the first was detected. All these studies document the importance of the MSS biocenosis, demonstrating the enormous potential of this subterranean habitat as a refuge for fauna, and constitute a good tool for the management of natural spaces.

## 2. Materials and Methods

### 2.1. Site

The sampling was conducted in the Sierra de Guadarrama National Park, located in the eastern half of the Central System in the Iberian Peninsula. It consists of an area 33,960 hectares, surrounded by a peripheral buffer zone of 62,687.26 hectares [12]. There are three mountainous axes (Siete Picos La Mujer Muerta, Montes Carpetanos, and Cuerda Larga) and an associated mountainous complex. The climate is Mediterranean, with marked continentality: winters are cold, and summers are cool and dry. A considerable variety of microclimates are derivate of the diverse topography of the mountains [13,14,15,16]. The study area is divided into three bioclimatic zones: supra-Mediterranean, oro-Mediterranean and cryo-Mediterranean [17,18]. The characteristics of these bioclimatic zones in the Sierra de Guadarrama are summarized in Ortuño et al. (2019) [19]. The snow also has importance, especially in areas that conserve snowfields for many months, such as in the cryo-Mediterranean and oro-Mediterranean scrub supra-forest zones.

### 2.2. Methodology

Thirty-three sampling points were established, located within the national park and its proximities, within the area declared a Peripheral Protection Area (Table 1, Figure 1): (a) five locations in Siete Picos and La Mujer Muerta, of which four had double SSDs (1 m and 0.5 m); (b) 15 localities in Montes Carpetanos, one of which had two TSPs installed and the remaining 14 an SSD of 1 m in length; (c) 12 localities with SSDs of 1 m in length, located in Cuerda Larga and its associated mountainous complex; and (d) two in Puerto de los Cotos-Puerto de Navacerrada. The methodology used for sampling was described in detail in Baquero et al. (2017) [8].

The sampling was performed by V. M. Ortuño, E. Ledesma, J. D. Gilgado, A. Jiménez Valverde, G. Pérez Suárez, and E. Baquero. Permits to collect samples were obtained from the appropriate authorities (General Directorate of Environment of the Community of Madrid and Territorial Service of the Environment of the Junta de Castilla y León). The traps (Table 1) were placed between 20 May 2015 and 9 July 2015, and the first series of samples was obtained first between 17 September 2015 and 6 November 2015, and then until dates close to May of 2016.

After the preliminary triage to separate the Collembola from the other taxons within the SSDs, some specimens of each OTU were selected and mounted in Hoyer’s medium for observation under a compound microscope in a phase contrast and DIC. Some specimens were cleared in Nesbitt’s fluid. The remaining samples were stored in 70% ethyl alcohol.

The macrochaetotaxy for *Pseudosinella* follows Gisin and Da Gama (1969), Szeptycki (1979), Soto Adames (2010), and Mateos (2008) [20,21,22,23]. The characters defined by Christiansen for *Pseudosinella*, used in a Delta key by Christiansen in Jordana et al. (2018) [24], were used for identification and descriptions. The equivalence between the notation proposed by Gisin (1965, 1967a, b) [25,26,27] and the AMS system *sensu* Soto-Adames 2010 [22] is given in Baquero et al. 2020 [28].

The abbreviations used are: a.s.l.—above sea level; Abd—abdomen or abdominal segment I–VI; accp—accessory posterior row sensillum; al—anterolateral s-chaeta; am—anteromedial s-chaeta; Ant—antennal or antenna/ae; Mc—Macrochaeta/ae; mes—mesochaeta; mic—microchaeta; PAO—Postantennal organ; pse—pseudopore; SSD—subterranean sampling devices; Th—thorax or thoracic segments II–III. Institutions: CAM—Comunidad de Madrid; JCL—Junta de Castilla y León; MNCN—Museo Nacional de Ciencias Naturales (CSIC), Madrid; National oMZNA—Museum of Zoology at the University of Navarra, Pamplona, Spain.

## 3. Results

### 3.1. Faunistic

The data for the taxon Collembola obtained from the first period were: 42,735 specimens, 31 genera and 65 species (16 new species). In this period, the most representative genus was *Orchesella*, represented by two new species (Baquero et al. 2017) [8]. In the second period, catches were significantly lower: 20,098 specimens, two genera were added (*Caprainea* and *Ceratophysella*), and 13 species not caught in the initial period. Twenty-six of the species captured in the first period were not captured in this one.

If the taxonomic groups lower than Collembola (Orders and Families) are considered (only those that provide sufficient abundance to allow a comparison to be made) it can be said that: Entomobryomorpha specimens decrease in the second period; Poduromorpha behaves differently depending on the family, with the Hypogastruridae decreasing and the Onychiuridae increasing; Symphypleona, in general, increases considerably, although there is a family (Sminthuridae) that suffers an appreciable decline.

Given that the traps had been in the field for a period of six months, and the specimens deteriorate over time, especially if they are not submerged in propylene glycol promptly, some identifications refer to specimens that had fallen into the trap close to its collection date.

### 3.2. Taxonomy

Class Collembola Lubbock, 1870 [29].

Order Poduromorpha Börner, 1913 [30], *sensu* d’Haese 2002 [31].

Family Hypogastruridae Börner, 1906 [32].

#### 3.2.1. *Ceratophysella meridionalis* (Steiner, 1955) [33] Nom. Nov. (http://zoobank.org/983D6632-E866-4DF8-AD0D-0364BCDC7EA9, accessed on 25 October 2022)

Studied material: pitfall SSD, 0.5 m and 1 m deep (since 17.IX.2015), trap SSD-01, 26.V.2016, 4 ♀♀, 1 ♂ (slide 04) and 134 specimens in ethyl alcohol; trap SSD–03, two specimens on slides 01 and 02 each, and 146 specimens in ethyl alcohol; trap SSD–08, five specimens on slide 04; trap SSD–17, three specimens on slide 04, two on slide 09 and 232 in ethyl alcohol; SSD–21, three specimens on slide 06 and 84 in ethyl alcohol. All Ortuño et al. leg., deposited at MZNA. Four syntypes from MNCN (9722, Cat. Tipos), slides numbered as: MNCN_Ent 106474; MNCN_Ent 106475; MNCN_Ent 106480; and MNCN_Ent 106481.

##### Description

Body length 1.3–2.0 mm. Body colour violet in alive specimens. Granulation rather coarse, 12–14 granules between chaetae p_1_ on Abd V (Yosii’s parameter) (Yoshii 1962) [34]. Along the head and body there are areas with fine granulation.

Antennae (Figure 2A,B). Ant IV with a big exsertile apical vesicle (av), subapical organite (or), microsensillum (ms), and 18 cylindrical and curved sensilla, 14–16 chaetae with rounded tip, and 21 acuminate chaetae. Ant III-organ with two short curved (rod-like) and two long (external) sensilla, 12 chaetae with ciliated half distal part (dorsal area), and six smooth chaetae (ventral area). Eversible sac (es) between Ant III–IV present (Figure 3). Ant II with eight ciliated (distal half, dorso-lateral) and five smooth chaetae (ventral). Ant I with seven chaetae, six ciliated at its tip (dorso-lateral).

Head. Ocelli 8 + 8. Head dorsal with a conspicuous tubercle (corneola-like structure following Stach 1964) [35] on either side lateral to chaeta sd_1_. PAO approximately two times the diameter of an ocelli, into a crevice, with two long and two short vesicles, and with an accessory boss (area with fine secondary granulation) (Figure 4 and Figure 5).

Mandibles with 4 or 5 teeth (asymmetric). Labrum with 5, 5, 4 chaetae, four prelabral chaetae present. Head of maxilla of the *C. armata* type. Maxillary outer lobe with two sublobal hairs. Labium of the *C. armata* type, with five papillae (A–E) and six proximal chaetae. Guard chaetae a_1_, b_1_, d_2_, e_2_ and lateral processes (l.p.) with rounded tip. Guards b_2–4_, d_3–4_, and e_1–6_ as long sensilla.

Chaetotaxy. Differentiation of dorsal chaetae into micro-, meso- and macrochaetae quite distinct. Head spine-like chaetae absent. Cephalic chaetae d_2_, d_5_, v_2_, p_1_, p_4_, p_6_, g_1_, g_5_ and l_01_ as Mc. Dorsal chaetotaxy of B type (sensu Gisin 1947 [36], Bourgeois and Cassagnau 1972, and Babenko et al. 1994) [37,38]. Apparently, all chaetae were pointed and serrated, and conspicuous from medium length chaetae. Th I with m_1_, m_3_ and m_4_ (following Jordana et al. 1997) [39]. Th II–III with Mc p_2_ and p_3_ shifted forward, p_2_ and p_6_ as Mc, and p_4_ and m_6_ as sensorial chaetae (s); Th II with microsensillum (ms), and chaetae a_2_ as long as a_3_ (Figure 5). Abd I–II with p_2_, and p_7_ as Mc, and p_5_ as sensorial chaetae (s); Abd III with p_2_, p_3_, p_4_ and p_7_ as Mc, and p_5_ as sensorial chaetae (s); Abd IV with p_1_, p_4_ p_6_ and p_7_ as Mc, p_5_ as sensorial chaetae (s); Abd V with p_1-2_ and p_7_ as Mc, and p_3_ as sensorial chaetae (s); Abd VI with many Mc. Body s-chaetae were relatively long, but shorter than macrochaetae, only those on Abd V were as long as macrochaetae (Figure 6).

Tibiotarsi I, II, III with 19, 19, 18 chaetae, respectively, including a clavate tenent hair longer than the claw in each. Claws with a big inner tooth and two pairs of lateral teeth. Empodial appendage with broad basal lamella and apical filament reaching slightly below inner tooth (ratio empodial filament: inner edge of claw : 0.70) (Figure 7A). Ventral tube with 4 + 4 chaetae (two apical and two basal).

Retinaculum with 4 + 4 teeth. Furca well developed. Ratio dens/mucro: 1.79. Dens with medium size granulation and two prominent tubercles between the two more distal chaetae (variable shape between specimens); seven dorsal chaetae of which one of the basal is slightly longer. Mucro with a sharpening tip, and with clear outer lamella (variable shape between specimens) (Figure 7B).

Anal spines 0.8 times shorter than inner edge of claw, slightly curved, situated on long papillae the same length as the colorless spine (Figure 6).

##### Ecology

In this project, five samples appeared, three in the highest part of the Montes Carpetanos (SSD-08, SSD-17 and SSD-21) and two on the north-face of La Mujer Muerta (SSD-01 and SSD-03). It seems linked to higher and colder areas.

##### Remarks

In 1955, Steiner [33] described a new species of *Hypogastura* s. str. from specimens collected above Cercedilla (Sierra de Guadarrama, Madrid, Spain). The description of the species is well documented but only the furca, the claw, and the anal spine are represented. We have considered important a re-description of this species due to the original description missing some characteristics that have recently been revealed as necessary. For this, not only the specimens collected by us (7851 ex.) but also four syntypes from the MNCN (Madrid) have been studied in detail. The species belongs to the armata-group (group B), Abd IV with p_1_ as macrochaeta (Thibaud et al. 2004) [40] and subgroup B1 (Abd IV with chaeta p_3_) (Bourgeois and Cassagnau 1972) [37]. It also has an accessory tubercle near the eyes on both sides of the head, similarly to *C. sinensis* Stach, 1964 [35] described from China, and also *H. tooliki* Fjellberg, 1985 [41], from Alaska and Magadan (Russia). The first species, unlike *C. meridionalis*, has the following diagnostic characteristics: color brown; body chaetae smooth; seven dorsal sensilla on dorsal, and some hook-like sensillum on the ventral side of Ant IV; head accessory tubercles are smooth, resembling a corneola; tenent hair short; empodial appendage short; dens basal chaeta two times the length of the other; dens without distal tubercles; dens to mucro ratio almost three times; mucro boat-like (rounded tip), and outer lamella not very developed; anal spines rounded at the tip, slender and not curved (Babenko et al. 1994) [38]. The second one has fine and uniform integumentary granulation, a weak difference between the body chaetae, all of them short and acuminate, body chaetae, and sensilla of the same size as the body chaetae.

None of the descriptions coincide with the description made, in his day, by Steiner or with the one made here. There is a clear heterochaetosis, with macrochaetae and microchaetae, the latter between ⅔ and ½ of the length of the Mc, all (Mc and mic) ciliated or with serrations, except for the sensilla that are smooth and longer or as long as the Mc. The furca is characteristic and as described by Steiner (1955) but has many sensilla and sensory chaetae of different types on the antenna (Figure 7B); very different from Steiner’s description and from that drawn by Jordana and Arbea in 1997 (Fauna Ibérica). Between Ant III and IV, the antenna has an eversible sac (Figure 3), characteristic of *Ceratophysella*. The shape of the PAO in *Ceratophysella* species is usually elongated (not as a rosette), as is also the case in this species.

When studying the definitions of the genera of *Ceratophysella* and *Hypogastrura*, it is observed that all the characteristics considered have exceptions, except heterochaetosis. In addition, since this species has clear heterochaetosis, it is considered to belong to the genus *Ceratophysella*. It is likely that DNA techniques can shed light on the difficulties of assigning a species to each of the genera.

Order Entomobryomorpha Börner, 1913 [30], sensu Soto-Adames et al. 2008 [42].

Family Lepidocyrtidae Wahlgren, 1906 [43] sensu Zang et al., 2015 [44].

Subfamily Lepidocyrtinae Wahlgren, 1906 [43].

Genus *Pseudosinella* Schäffer, 1897 [45].

#### 3.2.2. *Pseudosinella impariciliata* Baquero and Jordana Sp. Nov. (http://zoobank.org/17F77B1C-1527-4D37-93AD-A1D95AD20476, accessed on 24 October 2022)

##### Type Locality

Corrales de la Majada Minguete, (30 T 4100 45166, 1818 m a.s.l.), Sierra de Guadarrama, Segovia, Spain.

##### Type Material

Holotype. ♀, trap SSD-02 (slide 09), 26.v.2016, pitfall SSD (since 5.x.2015), Ortuño et al. leg. Paratypes. Same data as holotype: three specimens on slide (02), one each on slides 02, 10 and 11, and 48 (approximately) in ethyl alcohol.

##### Etymology

The name refers to the asymmetry of the prelabral chaetae, the central ones ciliated and the lateral ones smooth.

##### Diagnosis

Blue body pigment, including antennae and first leg segments. Head with 6 + 6 eyes; A_0_, A_2_ and A_3_ as Mc; basomedian labial fields chaetae smooth; posterior labial row with M_1_, m_2_, R*, e, L_1_ and L_2_ Mc; three plus one anterior postlabial chaetae as ciliate Mc. Th II–III without Mc; Abd II with chaeta a_2p_ present, a_3_ forward from ‘as’ sensilla; a_2_ as mic, and m_3_ as ciliated Mc; Abd IV with three median mac (C_1_, B_5–6_), four ciliated mic behind anterior bothriotrichum (some fan-shaped) and bothriothrichal complex mic D_1p_ present; claw with three internal teeth: two basal and one unpaired; empodium acuminate.

##### Description

Body length up to 1.90 mm, head included, excluding antennae (holotype: 1.80 mm). Color: dark blue on the whole body. Scales absent on antennae and legs, present on ventral and dorsal head, thorax, and abdomen dorsally, and furcula (dorsal and ventrally).

Head. Antennal head ratio 1.5 (*n* = 3). Ant IV without apical bulb, apical organite and accessory sensilla as in Figure 8A. Ant III sense organ with two rod-shaped sensilla (Figure 8B). 6  +  6 eyes (A–D, F–G). Head dorsal chaetotaxy with A_0_, A_2_ and A_3_ as Mc, A_2a_ present, and 22 chaetae at An row (eight as Mc); interocular t and p chaetae present (p as Mc) (Figure 8C); 4/554 chaetae on labral margin, prelabral external smooth and internal ciliated; labral a, m, and p row smooth (Figure 8D). Labral papillae absent, or not visible. Maxillary palp bifurcate with three smooth sublobal chaetae. Labial papilla (l.p.) E with finger-shaped process not extending to the base of apical appendage (Figure 8E). Labial row with M_1_, m_2_, R*, e, L_1_ and L_2_ Mc (R half to two-thirds of M; one specimen with M_2_ ciliated, asymmetric). Postlabial chaetotaxy with 3 + 1 ciliated central Mc along the groove (Figure 8F).

Abdomen chaetotaxy (Figure 9 and Figure 10). Abd II: mi and ml chaetae over bothriotrichum (m_2_); a_2p_ (p) present as slightly ciliated mic; a_2_ (A) as mes; m_3_ (B) as Mc; ‘as’ over m_3_ and a_2_, and a_3_ far above a_2_; m_3e_ and p_4_ (q_1_ and q_2_) as smooth (at least p_4_) mic; lm and ll as slightly ciliated mic over bothriotrichum (a_5_); a_6_, m_4_, m_6_ and p_5_ as smooth mic; m_5_ as mes; the rest of the chaetae are not seen. Abd III: mi, ml and a_2_ as broadened ciliated mic (fan-shaped) over bothriotrichum (m_2_); ‘as’ before m_3_ that is apparently smooth; a_3_, m_4_ and p_3_ as mic (shape not seen), a_3_ not very up; im, li, lm and a_6_ as slightly ciliated pointed mic surrounding bothriotrichum (a_5_); em, am_6_, a_7_ and a_7′_ (additional) as mic under and near a_5_ bothriotrichum; pm_6_ and p_6_ as Mc with d_3_ between them; ‘ms’ (d_2_) between p_5_ and p_6_; m_7_ as mic and p_7a_ as mes; m_7_–m_9_, p_7_ and p_9_ not seen clearly. Abd IV with three median mac (C_1_, B_5–6_; ratio between C_1_-B_5_/B_5_-B_6_ near 1.00, n = 2), and 7 lateral mac (D_3_, E_2–4_, F_1–3_); T_5_ as mic, D_2_, E_4p_, F_3p_, T_6_ and T_7_ as mes; before T_2_ bothriotrichum four ciliated mic (a, m, s and D_1_) some fan-shaped; pi and pe as ciliated fan-shape mic.

Legs. The legs are without scales. Trochanteral organ with 12–17 close spine-like chaetae (Figure 11A). Claw with three teeth on inner edge: paired at 50% and unpaired at 75% with respect to the internal claw edge length; two lateral teeth at 10–20%, and basal dorsal. Empodium acuminate, all legs with serrated pe lamella (very minute serration on leg 3), other lamellae smooth (ae, ai, pi); claw:empodium ratio = 1:0.65. Tibiotarsus III distally with one inner smooth chaeta 0.60 longer than claw; tenent hairs capitate, smooth, and 0.70 shorter than claw (Figure 11B).

Furcula. Manubrium and dens with scales both dorsal and ventrally, and with the same length; manubrial plate with three internal and 14–16 external chaetae, and 2 pse (Figure 11C). Non-ringed area of dens 3.5–4 times the length of mucro, with subapical tooth a little smaller than apical tooth. (Figure 11D).

Macrochaetotaxy. Reduced formula (from Gisin 1965, 1967a, b) [25,26,27]: R_0_R_1_R_2_000/00/0101 + 2/s, pABq_1_q_2_, M_1_m_2_R*eL_1_L_2_ (* ½ to 2/3 of M; sometimes M_2_ ciliated, asymmetric). Details in Figure 8 and Figure 9.

##### Ecology

The species is scarcely distributed in the MSS of Sierra de Guadarrama since it has only been found at one of the sampling points. The sampling area, at 1818 m asl, in the oro-Mediterranean bioclimatic zone, has the important forest influence of *Pinus sylvestris* L. The rock substrate is orthogneisses (Vialette et al. 1987) [46], and its characteristics facilitate the fracture of this material, which allowed its fragmentation during glacial (Pedraza and Carrasco 2005) [47] and periglacial (Sanz 1986) [48] events, usually in the form of large and medium-sized scree. The MSS where SSD-2 was installed shows a profile with an accumulation of large rocks at higher levels, and as depths reach 1 m, the size of the rock is reduced. Infiltration of black earth rich in organic matter was observed, but the MSS conserves the subterranean tridimensional network of fissures and interstices.

##### Remarks

The species that share the traditional formula of Gisin are in addition to *P. gonzaloi* Baquero and Jordana, 2021, and *P. valverdei* Baquero and Jordana, 2021 [10]. *P. gonzaloi* has the sensilla of the Ant III sensory organ, being rod-like, all labral chaetae ciliated (at least on its half distal part), the eyes in other positions, chaeta E on ventral labial row ciliated, four teeth on internal or ventral claw, and a different number of chaetae on the manubrial plate (7–8). *P. valverdei* has only five eyes, prelabral chaetae smooth, L_1_ and L_2_ chaetae on ventral labial row smooth, and four teeth on the internal or ventral claw. Curiously, these two species have been described in the same mountain massif. This species could have gone unnoticed among the thousands of specimens of *Pseudosinella* and *Lepidocyrtus* found in the first period. Most of the specimens were very deteriorated after remaining between several days and six months in the propylene glycol of the traps.

## 4. General Discussion

The difference between the catches of the first period and the second, when comparing the species found, is striking. Many of the species well represented and in abundance during the period between May and October went from high numbers (tens of thousands of individuals) to only hundreds in the following period (October to May), and in some cases, even a few units or disappear (*Entomobrya ledesmai* Jordana and Baquero, 2021 and *Pseudosinella gonzaloi* Baquero and Jordana, 2021) (see Table 2). The opposite also happens: species little represented in the first period increase in their abundance in the second, going from hundreds to thousands, even in one case, from complete absence to being one of the most abundant species with thousands of specimens (*Lepidocyrtus labyrinthi* Baquero and Jordana, 2021 and *Hypogastrura papillata* Gisin, 1949 [49]). There is a case to comment on: *C.  meridionalis* Steiner, 1955 [33] was the most abundant species in the first period and seemed to be replaced by *H. papillata*. During the placement of the traps, a few weeks before the month of May, it was possible to see accumulations of this species on the surface, under the slabs of cattle excrement in the highland meadows. This indicates that it is present, at least during part of the year, on the surface of the ground. Finding it during the following months in the MSS confirms that it moves vertically on the ground during the year, and the fact that between May and October 621 specimens were captured in the traps indicates that during this period it prefers the surface.

This reasoning cannot be applied to other abundant species between May and October (*Orchesella colluvialis*, *Lepidocyrtus paralignorum*, *Entomobrya guadarramensis*, *E. ledesmai*) since they are not species detected on the surface at any time before. The decrease in their abundance between May and October must be due to other reasons; for example, biological cycles adapted to temperature or food availability.

No reference has been made to other papers that refer to karstic MSS because this biotope is granitic. The fauna found in this granitic MSS is mainly its own fauna, as has been shown in our previous works. This study indicates that the colluvial MSS has a particular species composition for the taxon Collembola, demonstrating that the MSS should not be considered a mere ecotone between the surface and the deep subterranean ecosystem, but rather as a perfectly differentiated habitat.

## Figures and Tables

**Figure 1 insects-13-01025-f001:**
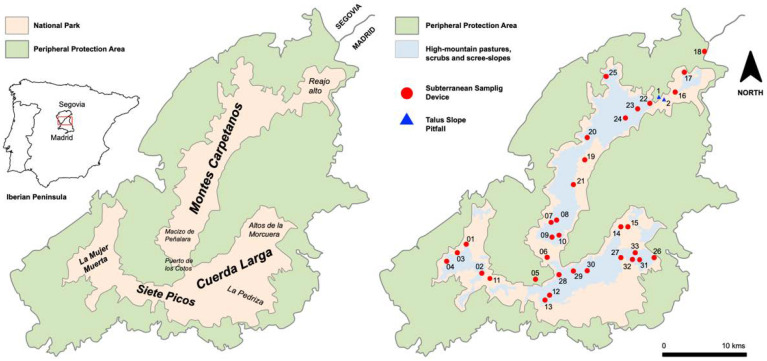
Maps of the Sierra de Guadarrama National Park. Basic orography (**left)**, and habitats that occupy greater surfaces (**right)**. The spatial location of the sampling points is indicated and numbered.

**Figure 2 insects-13-01025-f002:**
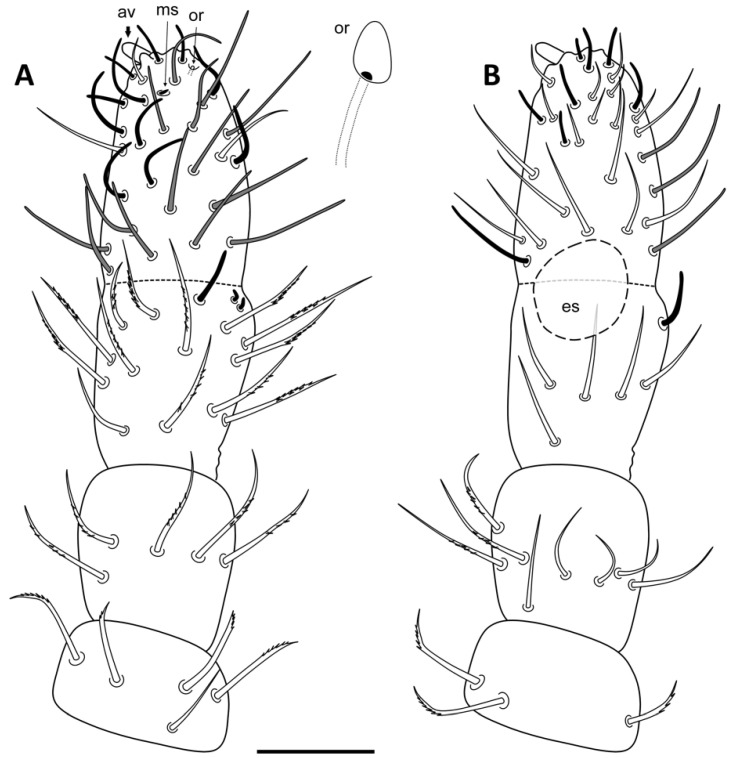
*Ceratophysella meridionalis*: antennal segments I to IV, dorsal view at left (**A**); ventral view at right (**B**) (av, apical vesicle; ms, microsensillum; or, subapical organite; es, eversible sac) (scale bar, 0.05 mm).

**Figure 3 insects-13-01025-f003:**
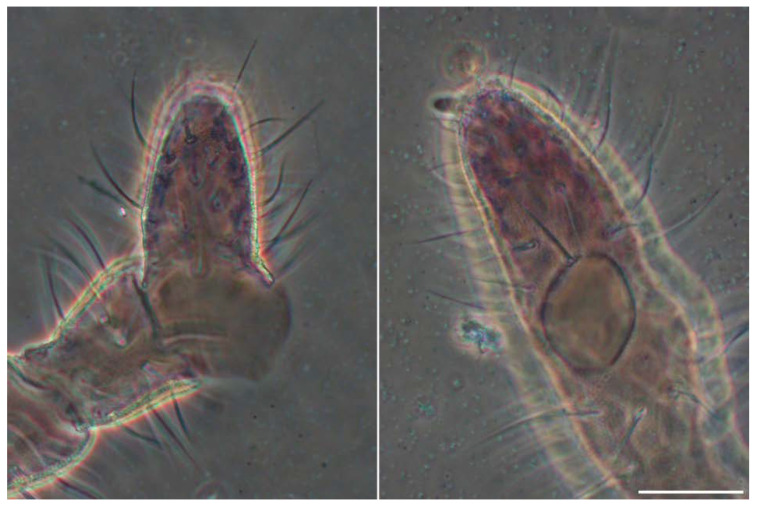
*Ceratophysella meridionalis*: eversible sac, two different views (scale bar, 0.05 mm).

**Figure 4 insects-13-01025-f004:**
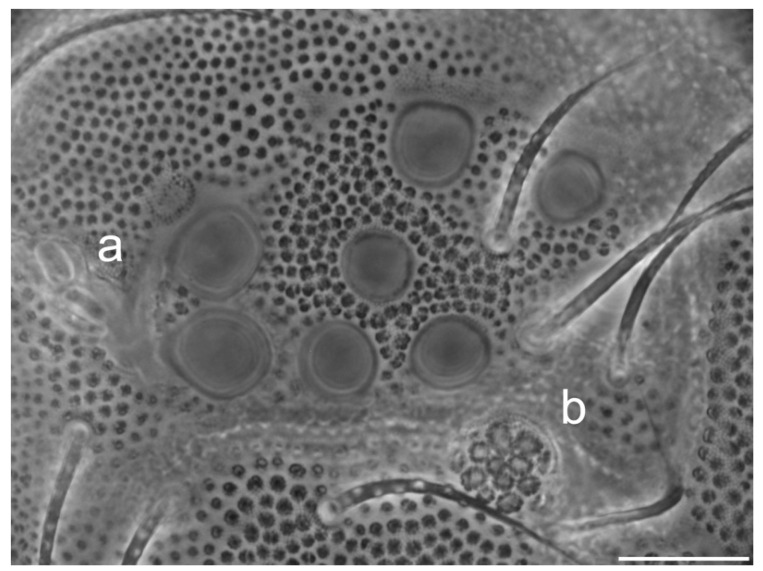
*Ceratophysella meridionalis*: ocular area, with PAO at left (**a**) and tubercle at right (**b**) (scale bar, 0.02 mm).

**Figure 5 insects-13-01025-f005:**
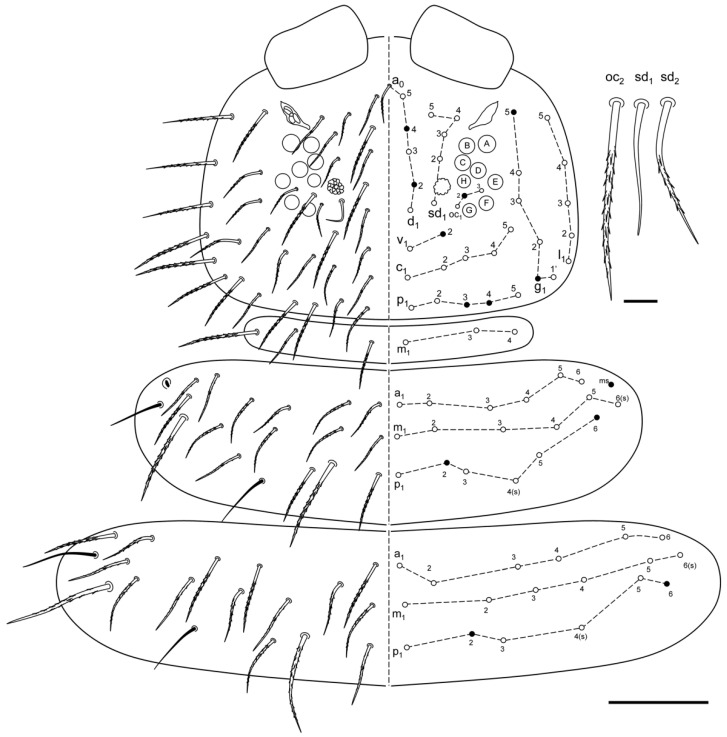
*Ceratophysella meridionalis*: head and Th I–III, with detail on some of the chaetae (scale bars: head and thoracic tergites, 0.1 mm; chaetae: 0.01 mm).

**Figure 6 insects-13-01025-f006:**
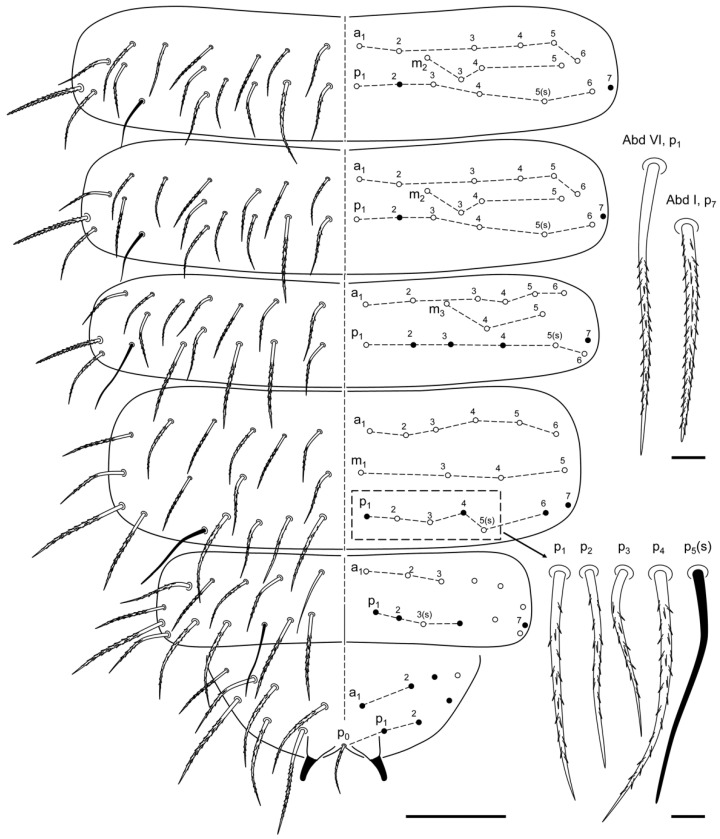
*Ceratophysella meridionalis*: Abd I–VI, with detail on some of the chaetae (scale bars: abdominal tergites, 0.1 mm; chaetae, 0.02 mm).

**Figure 7 insects-13-01025-f007:**
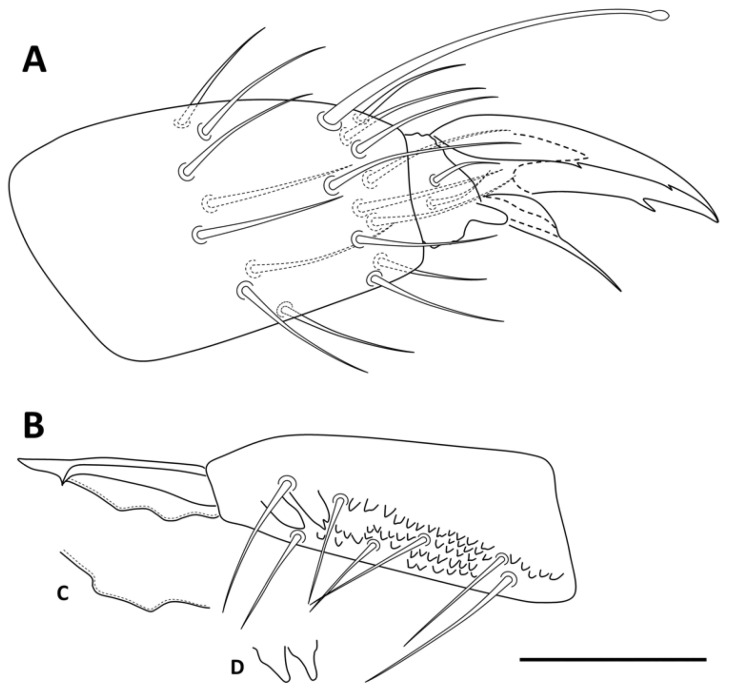
*Ceratophysella meridionalis*: (**A**) tibiotarsi; (**B**) dentes and mucro, with detail of the variations in the lamellae of mucro shape (**C**) and tubercle shaped teeth (**D**) (scale bar, 0.05 mm).

**Figure 8 insects-13-01025-f008:**
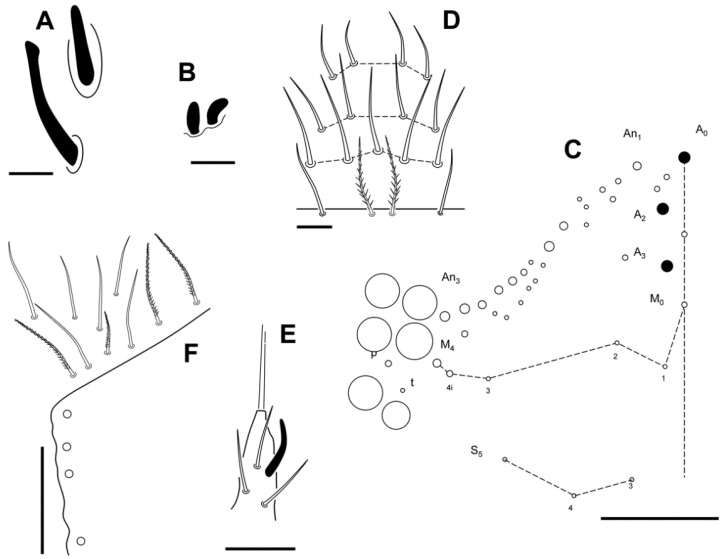
*Pseudosinella impariciliata* Baquero and Jordana sp. nov.: (**A**) Abd IV subapical organite, and accessory chaetae; (**B**) rod-shaped sensilla of the Ant III sensory organ; (**C**) dorsal head chaetotaxy; (**D**) prelabral and labral chaetae; (**E**) labial papilla E; (**F**) post-labial chaetae (scale bars: (**A**,**B**) 0.001 mm; (**C**,**F**) 0.05 mm; (**D**,**E**) 0.01 mm).

**Figure 9 insects-13-01025-f009:**
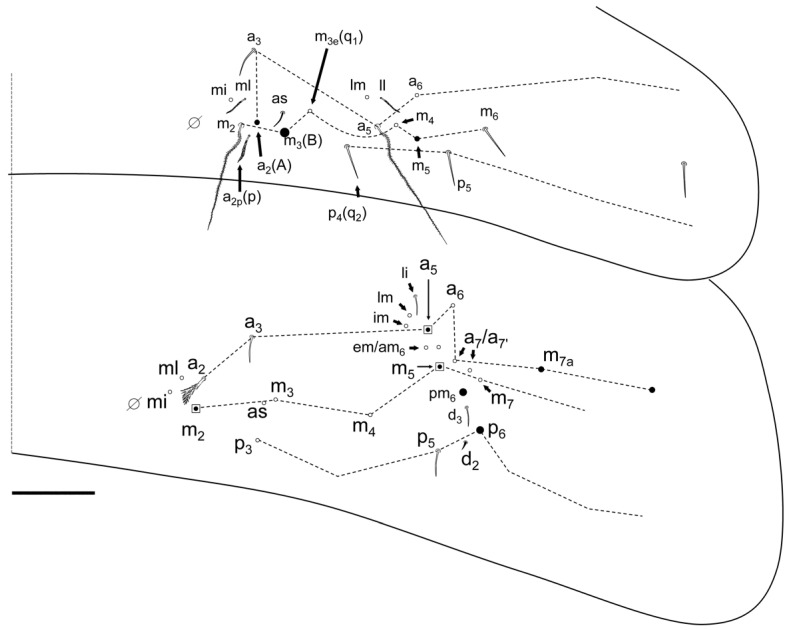
*Pseudosinella impariciliata* Baquero and Jordana sp. nov.: Abd II–III chaetotaxy (scale bar, 0.05 mm).

**Figure 10 insects-13-01025-f010:**
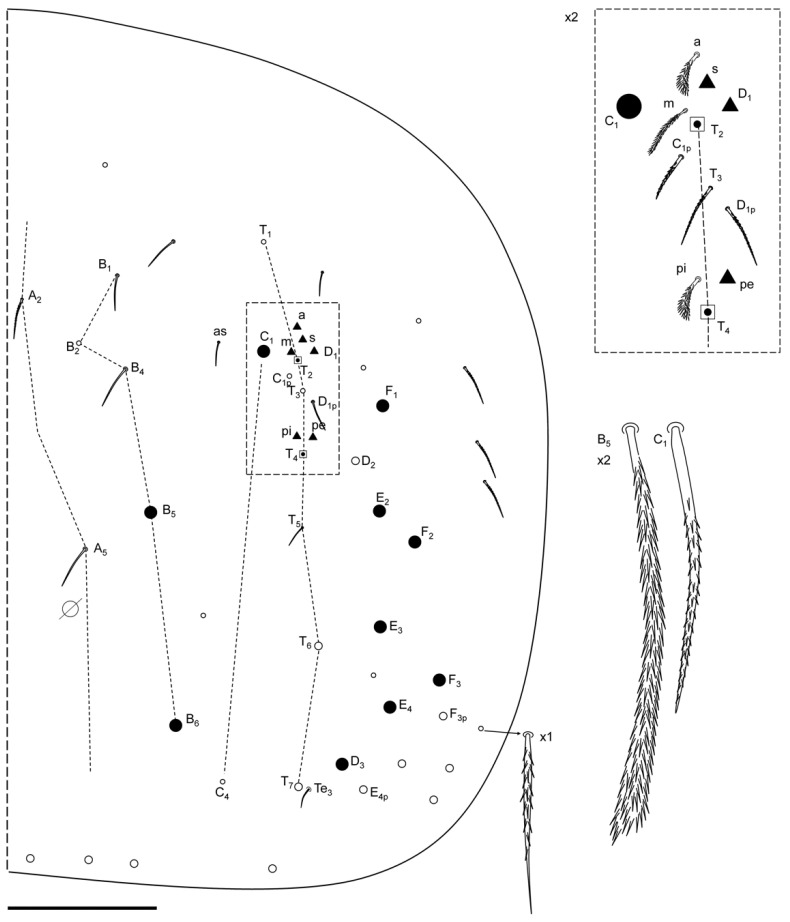
*Pseudosinella impariciliata* Baquero and Jordana sp. nov.: Abd IV chaetotaxy (scale bar, 0.1 mm).

**Figure 11 insects-13-01025-f011:**
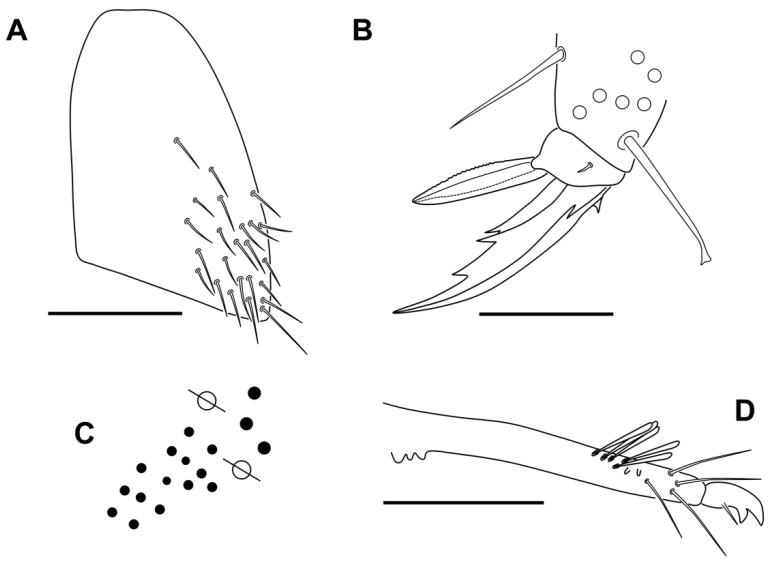
*Pseudosinella impariciliata* Baquero and Jordana sp. nov.: (**A**) trochanteral organ; (**B**) claw of leg III; (**C**) pseudopores and chaetae of manubrial plate; (**D**) distal area of dentes and mucro (scale bar, 0.05 mm).

**Table 1 insects-13-01025-t001:** Location of the traps in the mountain areas of the Sierra de Guadarrama. Values: depth: meter; UTM coordinates: 100 × 100 m; altitude: m a.s.l. Abbreviations: SSD, subterranean sampling device.

Mountain Areas of the Sierra de Guadarrama	Code	Depth (m)	UTM Coordinates (100 × 100 m)	Altitude	Toponymy/Province	Date of Trap Installation	Date of Trap Recovery	Orientation
Siete Picos-La Mujer Muerta	SSD-01	1	30 T 4081 45,204	1606	Cancho del Río Peces/Segovia	20 May 2015	17 September 2015	North
	SSD-01 (0.5)	0.5						
	SSD-02	1	30 T 4100 45,166	1818	Corrales de la Majada Minguete/Segovia	20 May 2015	17 September 2015	Northeast
	SSD-02 (0.5)	0.5						
	SSD-03	1	30 T 4068 45,192	1622	Umbría de la Mujer Muerta/Segovia	21 May 2015	17 September 2015	North
	SSD-03 (0.5)	0.5						
	SSD-04	1	30 T 4056 45,181	1685	Majada Conejo/Segovia	21 May 2015	17 September 2015	Northwest
	SSD-04 (0.5)	0.5						
	SSD-11	1	30 T 4108 45,161	1876	Cerro Ventoso/Madrid	9 June 2015	17 September 2015	East
Puerto de los Cotos-Puerto de Navacerrada	SSD-05	1	30 T 4166 45,159	1923	Arroyo Seco/Segovia	27 May 2015	22 September 2015	Northwest
	SSD-06	1	30 T 4179 45,185	1787	La Pedriza/Segovia	27 May 2015	22 September 2015	Northwest
Montes Carpetanos	SSD-07	1	30 T 4185 45,229	1994	Majada Hambrienta/Segovia	2 June 2015	17 September 2015	Northwest
	SSD-08	1	30 T 4190 45,231	2071	Majada Aranguez/Segovia	2 June 2015	17 September 2015	Northwest
	SSD-09	1	30 T 4187 45,211	2208	Dos Hermanas/Madrid	3 June 2015	5 October 2015	East
	SSD-10	1	30 T 4191 45,213	2049	Hoya de la Laguna Grande/Madrid	3 June 2015	5 October 2015	East
	SSD-16	1	30 T 4334 45,389	1956	Las Revueltas-Los Horcos/Segovia	23 June 2015	7 October 2015	West
	SSD-17	1	30 T 4347 45,414	1976	Peña del Buitre/Segovia	23 June 2015	7 October 2015	Northwest
	SSD-18	1	30 T 4373 45,438	1885	Los Loberos/Segovia	23 June 2015	7 October 2015	Northwest
	SSD-19	1	30 T 4224 45,307	1866	La Gelecha-La Flecha/Madrid	24 June 2015	6 October 2015	Southeast
	SSD-20	1	30 T 4226 45,332	1937	Cerro de Navahonda/Segovia	24 June 2015	6 October 2015	Northeast
	SSD-21	1	30 T 4211 45,274	1891	El Paredón/Madrid	24 June 2015	6 October 2015	Northeast
	SSD-22	1	30 T 4304 45,376	1995	Alto del Puerto/Segovia	24 June 2015	22 September 2015	North
	SSD-23	1	30 T 4288 45,367	2144	Circo del Pico Nevero/Madrid	25 June 2015	6 October 2015	Southeast
	SSD-24	1	30 T 4274 45,357	2042	Peñacabra/Madrid	25 June 2015	22 October 2015	East
	SSD-25	1	30 T 4249 45,407	1731	Arroyo del Charco (La Cepa)/Segovia	2 July 2015	22 October 2015	Northwest
	TSP-1	0.8	30 T 4314 45,376	1780	Puerto de Navafría/Segovia	24 June 2015	22 September 2015	North
	TSP-2	0.8	30 T 4314 45,376	1780	Puerto de Navafría/Segovia	24 June 2015	22 September 2015	North
Cuerda Larga and Associated Mountainous complex	SSD-12	1	30 T 4180 45,138	2102	Collado del Piornal/Madrid	9 June 2015	22 September 2015	North
	SSD-13	1	30 T 4179 45,135	2113	Los Almorchones-Las Buitreras/Madrid	10 June 2015	22 September 2015	Southwest
	SSD-14	1	30 T 4274 45,224	1406	El Purgatorio/Madrid	18 June 2015	5 October 2015	West
	SSD-15	1	30 T 4273 45,224	1375	Hueco de los Ángeles/Madrid	18 June 2015	5 October 2015	Northeast
	SSD-26	1	30 T 4309 45,186	1890	La Najarra-Cuatro Calles/Madrid	2 July 2015	30 October 2015	East
	SSD-27	1	30 T 4270 45,185	2101	Bailaderos/Madrid	2 July 2015	30 October 2015	North
	SSD-28	1	30 T 4193 45,164	2156	Collado de Valdemartín/Madrid	3 July 2015	6 November 2015	North
	SSD-29	1	30 T 4211 45,168	2301	Cabeza de Hierro Mayor Menor/Madrid	3 July 2015	6 November 2015	Crest
	SSD-30	1	30 T 4227 45,170	2233	Collado de Peña Vaqueros (Loma de Pandasco)/Madrid	3 July 2015	6 November 2015	Crest
	SSD-31	1	30 T 4288 45,184	1946	Collado de la Najarra/Madrid	9 July 2015	22 October 2015	North
	SSD-32	1	30 T 4285 45,187	1948	Arroyo de La Najarra/Madrid	9 July 2015	22 October 2015	Northeast
	SSD-33	1	30 T 4286 45,188	1819	Arroyo de La Najarra/Madrid	9 July 2015	22 October 2015	North

**Table 2 insects-13-01025-t002:** Total list of species found in the sampling, in the two sessions, ordered by abundance for the first extraction from the traps. The numbers are the total number of specimens for each species.

Species	May–October	October–May	Total
*Ceratophysella meridionalis* (Steiner, 1955) *nom. nov.*	7230	621	7851
** Orchesella colluvialis* Jordana and Baquero, 2017	7150	575	7725
** Lepidocyrtus paralignorum* Baquero and Jordana, 2021	6480	8	6488
** Entomobrya guadarramensis* Jordana and Baquero, 2021	5455	202	5657
** Entomobrya ledesmai* Jordana and Baquero, 2021	4109	0	4109
** Pseudosinella valverdei* Baquero and Jordana, 2021	2505	1461	3966
** Orchesella mesovoides* Baquero and Jordana, 2017	1982	178	2160
** Pygmarrhopalites custodum* Baquero and Jordana, 2021	1531	4801	6332
*Heteromurus major* (Moniez, 1889)	1302	2435	3737
*Xenylla schillei* Börner, 1903	1066	2	1068
** Pseudosinella gonzaloi* Baquero and Jordana, 2021	906	0	906
*Xenylla maritima* Tullberg, 1869	615	26	641
** Pachyotoma penalarensis* Baquero and Jordana, 2021	538	510	1048
*Lepidocyrtus tellecheae* Arbea and Jordana, 1990	347	3	350
** Lepidocyrtus labyrinthi* Baquero and Jordana, 2021	280	2687	2967
*Gisinurus malatestai* Dallai, 1970	228	2	230
*Protaphorura subparallata* (Selga, 1962)	227	3410	3637
*Tetracanthella proxima* Steiner, 1955	215	9	224
*Pseudosinella simoni* Jordana and Baquero, 2007	137	7	144
*Pseudisotoma monochaeta* (Kos, 1942)	61	355	416
*Xenylla lotharingiae* Thibaud, 1963	45	0	45
** Allacma cryptica* Baquero and Jordana, 2021	45	0	45
*Hypogastrura affinis* (Steiner, 1955)	38	0	38
*Protaphorura valsainiensis* (Acón, 1981)	37	0	37
*Entomobrya albocincta* (Templeton, 1836)	24	0	24
** Lepidocyrtus purgatori* Baquero and Jordana, 2021	22	0	22
*Pygmarrhopalites elegans* Cassagnau and Delamare Deboutteville, 1953	27	0	27
*Lepidocyrtus lusitanicus nigrus* Simón-Benito, 2007	19	0	19
*Protaphorura florae* Simón-Benito and Luciáñez, 1994	18	0	18
** Schaefferia sendrai* Jordana and Baquero 2020	12	5	17
*Protaphorura spinoidea* (Steiner, 1955)	11	0	11
*Xenylla nitida* Tullberg, 1971	9	0	9
*Arrhopalites caecus* (Tullberg, 1871)	9	0	9
** Friesea ortugnoi* Jordana and Baquero, 2020	8	1	9
*Megalothorax minimus* Willem, 1909	7	0	7
*Folsomia trisetata* Jordana and Ardanaz, 1881	5	3	8
*Sminthurinus gisini* Gama 1961	5	0	5
*Sphaeridia pumilis* (Krausbauer, 1898)	5	0	5
*Isotomurus* sp.	4	2	6
*Xenylla mediterranea* Gama, 1964	3	0	3
*Dicyrtomina minuta* (Fabricius, 1783)	3	0	3
*Folsomides portucalensis* Gama, 1961	2	0	2
*Uzelia kuehnelti* Cassagnau, 1954	2	0	2
*Hypogastrura conflictiva* Jordana and Arbea, 1992	2	0	2
*Bilobella aurantiaca* (Caroli, 1912)	2	6	8
*Entomobrya nicoleti* (Lubbock, 1870)	1	0	1
*Parisotoma notabilis* (Schäffer, 1896)	1	1	2
*Tetracanthella orbaicetensis* Cassagnau, 1959	1	0	1
*Mesogastrura ojcoviensis* (Stach, 1919)	1	1	2
*Pseudachorutes palmiensis* Börner, 1903	1	3	4
*Fasciosminthurus* sp.	1	0	1
*Sminthurides* sp.	1	0	1
*Hypogastrura papillata* Gisin, 1949	0	2286	2286
*Ceratophysella* cf. *bengtssoni* (Ågren, 1904)	0	363	363
** Pseudosinella impariciliata* Baquero and Jordana, 2021	0	54	54
*Sminthurinus aureus*	0	49	49
*Ceratophysella succinea* (Gisin, 1949)	0	11	11
*Ceratophysella engadinensis*(Gisin, 1949)	0	10	10
*Pseudosinella templadoi* Simón and Selga, 1977	0	6	6
*Caprainea* sp.	0	1	1
*Ceratophysella* cf. *gibbosa* (Bagnall, 1940)	0	1	1
*Folsomia manolachei* Bagnall, 1939	0	1	1
*Folsomia quadrioculata* (Tullberg, 1871)	0	1	1
*Protaphorura glebata* (Gisin, 1952)	0	1	1
	42,735	20,098	62,833

*  Indicates that it is a species described as new in relation to the Project and the publications derived from it. The species not identified to species level were juveniles or damaged specimens.

## Data Availability

All data is contained within the article. Collected specimens have been deposited in the collection of the Museum of Zoology, Department of Environmental Biology, University of Navarra (MZNA).

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
