# Peer review of "Collembola from Two Samplings in the MSS of the Sierra de Guadarrama National Park in Two Different Seasons, with Description of a New Species"

_insects, 2022, doi:10.3390/insects13111025_

Round 1

Reviewer 1 Report

The reviewed work "Collembola from two samplings in the MSS of the Sierra de Guadarrama National Park in two different seasons, with description of two new species." by Baquero E. & Jordana R. presents the results of:

- original faunistic research on Collembola of colluvial mesovoid shallow substratum in Sierra de Guadarrama (Spain),

- original taxonomic studies on Ceratophysella granuloculata Jordana & Baquero sp. nov. and Pseudosinella impariciliata Baquero & Jordana sp. nov.

The work provides new valuable information on springtails found in the same traps but in different periods of the year and provides modern descriptions of species mentioned above. In my opinion, the work falls within the scope of Insects and is suitable for publication. However, some modifications are necessary.

First of all, Authors should:

1. Re-examine the taxonomic status of Ceratophysella granuloculata sp. nov. Considering pointed mucro, presence of dorsal cylindrical sensilla below microsensillum on Ant. IV, clavate hairs on tibiotarsi, setae m2 on Th. II-III and short anal spines it is rather Hypogastrura. Moreover, it appears identical to Hypogastrura meridionalis Steiner, 1955 (described from Sierra de Guadarrama and numerous in and outside the MSS). Especially if we take into account the description contained in the work of Dallai & Ferrari (1970), who studied specimens from Italy, but also paratypes.

2. More precisely justify the conclusion "This study indicates that the colluvial MSS has a particular species composition for the taxon Collembola, demonstrating that the MSS should not be considered a mere ecotone between the surface and the deep subterranean ecosystem, but rather as a perfectly differentiated habitat." (lines 344-346). For example, I propose to add a column with ecological preferences of the species listed in Table 2, add a separate table presenting the total number of identified briophilic, soil species… etc., and make a detailed analysis of the share of individual ecological groups. This may also help explain the differences between the results obtained in the two study periods (V-X vs XI-IV).

In addition, I noticed some minor errors:

1. There is no information on the presence/absence of microsensillum on Ant. IV in the description of C. granuloculata sp. nov.

2. In Fig. 1 there is no explanation what the structure is marked with the symbol A '. In Fig 1B it is not explained what is a broken line circle (possibly an eversible sac?).

3. In Fig. 3, setae m4 on Th. II-III should be marked as m5, and setae m5 as m6 and m6 as m7.

4. In the diagnosis of C. granuloculata sp. nov. is the following information: "Ant IV with a big apical exertile vesicle with a series of accessory smaller bulges, a ventral sensory file with 10–11 broad peg-like s-chaetae." (lines 135-137). There is no such information in the description.

Author Response

Answer to referee 1

1) Re-examine the taxonomic status of Ceratophysella granuloculata sp. nov. Considering pointed mucro, presence of dorsal cylindrical sensilla below microsensillum on Ant. IV, clavate hairs on tibiotarsi, setae m2 on Th. II-III and short anal spines it is rather Hypogastrura. Moreover, it appears identical to Hypogastrura meridionalis Steiner, 1955 (described from Sierra de Guadarrama and numerous in and outside the MSS). Especially if we take into account the description contained in the work of Dallai & Ferrari (1970), who studied specimens from Italy, but also paratypes.

Answer: We greatly appreciate the reviewer's contribution in this regard, since the species is indeed Ceratophysella meridionalis. We have requested on loan and studied four specimens of the type series of Hypogastrura meridionalis Steiner, 1955. It has allowed us to confirm the identity, and to observe some characters not mentioned by Steiner, which went unnoticed in the previous work (in which we quoted H. meridionalis). . We have made a redescription of the species, changing its gender due to the presence of the observed characters: heteochaetosis, presence of exsertile vesicle between antennal segments 3 and 4, and PAO shape.

2) More precisely justify the conclusion "This study indicates that the colluvial MSS has a particular species composition for the taxon Collembola, demonstrating that the MSS should not be considered a mere ecotone between the surface and the deep subterranean ecosystem, but rather as a perfectly differentiated habitat." (lines 344-346). For example, I propose to add a column with ecological preferences of the species listed in Table 2, add a separate table presenting the total number of identified briophilic, soil species… etc., and make a detailed analysis of the share of individual ecological groups. This may also help explain the differences between the results obtained in the two study periods (V-X vs XI-IV).

We believe that it would be very difficult to make the change to the table that the reviewer suggests. The information it mentions is very scarce for the species included, and furthermore the most abundant species have only been found in this study (all except one, which is from the surface and depth).

3) There is no information on the presence/absence of microsensillum on Ant. IV in the description of C. granuloculata sp. nov.

Answer: done

4) In Fig. 1 there is no explanation what the structure is marked with the symbol A '. In Fig 1B it is not explained what is a broken line circle (possibly an eversible sac?).

Answer: done

5) In Fig. 3, setae m4 on Th. II-III should be marked as m5, and setae m5 as m6 and m6 as m7.

Answer: Chaetotaxy follows the work of Jordana and Arbea (1997: 62-63).

6) In the diagnosis of C. granuloculata sp. nov. is the following information: "Ant IV with a big apical exertile vesicle with a series of accessory smaller bulges, a ventral sensory file with 10–11 broad peg-like s-chaetae." (lines 135-137). There is no such information in the description.

Reviewer 2 Report

Dear Editor and Authors,

I recognise here two parts/interests. One, which includes faunistic and diversity data of Collembola in MSS environment for which more marked conservational approach should be included and classical morphology with the quality level. There is rather descriptive  faunistic data concerning the first part of ms, less related to ecology of Collembola inhabiting MSS in different sampling periods, while  the taxonomic descriptive part is well done and does not raise major problem. My friendly comment to authors, there is a need to improve „Introduction“. I especially like the very accurate background information on research issue, there is important to introduce more details for readers. Please, formulate aims clearly in this part. More critical approach is needed, some useful works are absent in this section. Confrontation of authors results with  other comparative studies and also more details to MSS fauna from conservation point of the view is expected in the section „Discussion“. My remarks, comments and suggestions are also included  in Pdf.

Author Response

Contestación al referee 2

1) This paper is the fifth dedicated exclusively to the taxon Collembola in relation with this study (Redundant information for readers. Please, remove this part.)

Answer: It does not seem redundant information because it has not been said before anywhere in the text.

2) Previous studies were focused on ecology [3–4], or 35 faunal aspects [5–7] (I have no doubt, that authors are well informed about fauna from MSS habitat, please, can you add several recent references relevant to this statement, e.g. Rendoš, M.,  Raschmanová, N., Kováč, Ľ., Miklisová, D., Mock, A., Ľuptáčik, P., 2016: Organic carbon content and temperature as substantial factors affecting diversity and vertical distribution of Collembola on forested scree slopes. European Journal of Soil Biology, 75: 180–187; Jureková, N., Raschmanová, N., Miklisová, D., Kováč, Ľ., 2021: Mesofauna at the soil-scree interface in a deep karst environment. Diversity, 13(6), 242.

Answer: We must remember that Jurekova's paper refers to a karstic environment, very different from that of the Sierra de Guadarrama, which is Granitic. No reference has been made to other papers referring to karstic MSS because this biotope is granitic. In a karstic environment, there are caves and the scree-slope can be an intermediate place between the caves and the outside environment. The springtails that live in caves are not really from caves but from the infractions of the karstic terrain, which by definition is very broken. There are no caves in terrains such as the MSS de Guadarrama, and the only place far from the surface is the bottom of about two meters of the scree-slope. The fauna found in it is mainly its own fauna, as has been shown in our previous works.

In relation to this paper (Rendos), we have not included these factors (coal and temperature) in our studies nor have we made any relation to them.

We have included an additional paragraph in the General Discussion to make it clear what the context of our study is. We appreciate the remark made.

3) The first initial period...ok, please to give information about second period, only reference about sampling desiguan, this is insufficent for reader.

Answer. As has been said in methodology, in this second period the same traps have been used as in the first period and with the same sampling system. In addition, there is expanded information in the earlier works cited in this one.

4) 16 new species, it is unclear for me...

Answer: “The data for the taxon Collembola obtained from the first period were: 42 735 90 specimens, 31 genera and 65 species (16 new species)”. This same sentence explains that of the 65 species found in the first period, 16 were new to science, despite the fact that this mountain range has been sampled by Steiner, Bonet, Selga, Simón, and Luciañez. This indicates that there is its own underground fauna that lives in the MSS and is not found on the surface.

5) More critical discussion is needed, confrontation of authors results with other comparative studies, which inform us about "MSS fauna", also Collembola, in two periods including also the first initial sampling period, e.g. Mock A, Šašková T, Raschmanová N, Jászay T, Ľuptáčik P, Rendoš M, Tajovský K, Jászayová A. (2015) An introductory study of subterranean communities of invertebrates in forested talus habitats in southern Slovakia. Acta Societatis Zoologicae Bohemicae 79: 243–256

Answer: We are very grateful for the information received, which was already known. The biotopes sampled in our work have no soil, they are broken rocks from the erosion of high mountains (colluvial) and can be a refuge for part of the Fauna found, as has already been described in the previous papers.

In the present paper, we show the results of a second sampling period that has yielded unexpected results in the number of springtail species. Some species are on the surface, always in small numbers, while others, the most abundant, belong to the sampled biotope and have never been found outside (this is true for all the abundant species except for one, Ceratophysella meridionalis, found both on the surface and in the MSS.

6) In my opinion,here it is redundant information, maybe more relevant for "Author contribution"

“The sampling was performed by V. M. Ortuño, E. Ledesma, J. D. Gilgado, A. Jiménez 63 Valverde, G. Pérez Suárez and E. Baquero” Answer: Most of these authors do not sign the paper; it is not possible to mention them in "Author Contributions". That is why we have decided to mention them here.

7) “marked text in "Results" includes data of Table 2. Please, insert it here. It is difficult for reader, that this Table is visible latter... in Discussion.”

Anwer: Thanks. We have changed the location of the table where it is suggested to us

8) “Please, add species authority for Collembola species”. Table 2.

Answer: done.

Round 2

Reviewer 1 Report

I accept the authors' responses to my comments. The reviewed manuscript " Collembola from two samplings in the MSS of the Sierra de Guadarrama National Park in two different seasons, with description of a new species." by Baquero E. & Jordana R. in its current form is almost ready to be published . However, some minor modifications are still necessary:

1. Information in the description of C. meridionalis regarding postantennal organ, clavate tenent hair and papillae of anal spines has been unnecessarily duplicated:

- (lines 167-168) „Postantennal organ 1.6 times as large as single ocellus with four lobes of which the anterior pair is narrower than the posterior.” and (lines 170-172) „PAO two times the diameter of an ocelli, into a crevice, with two long and two short vesicles, and with an accessory boss (area with fine secondary granulation) next to PAO (Figs 2–3–4).”

- (lines 199-200) „Tibiotarsi I, II, III with 19, 19, 18 chaetae respectively, including a clavate tenent hair in each.” and (lines 200-201) „Tennent hair longer than the claw and clavate.”

- (lines 214 215) „Anal spines 0.8 times shorter as inner edge of claw, slightly curved, situated on large basal papillae situated on long papillae the same length as the spine, colourless (Fig. 45).”

2. Line 378: now is „H. meridionalis”, should be „C. meridionalis”

3. Line 384: now is „32 specimens”, should be „621 specimens”

Author Response

Reviewer 1 said: 

I accept the authors' responses to my comments. The reviewed manuscript " Collembola from two samplings in the MSS of the Sierra de Guadarrama National Park in two different seasons, with description of a new species." by Baquero E. & Jordana R. in its current form is almost ready to be published . However, some minor modifications are still necessary:

  1. Information in the description of C. meridionalis regarding postantennal organ, clavate tenent hair and papillae of anal spines has been unnecessarily duplicated:

- (lines 167-168) „Postantennal organ 1.6 times as large as single ocellus with four lobes of which the anterior pair is narrower than the posterior.” and (lines 170-172) „PAO two times the diameter of an ocelli, into a crevice, with two long and two short vesicles, and with an accessory boss (area with fine secondary granulation) next to PAO (Figs 2–3–4).”

Answer: Done.

- (lines 199-200) „Tibiotarsi I, II, III with 19, 19, 18 chaetae respectively, including a clavate tenent hair in each.” and (lines 200-201) „Tennent hair longer than the claw and clavate.”

Answer: Done.

- (lines 214 215) „Anal spines 0.8 times shorter as inner edge of claw, slightly curved, situated on large basal papillae situated on long papillae the same length as the spine, colourless (Fig. 45).”

Answer: Done.

  1. Line 378: now is „H. meridionalis”, should be „C. meridionalis”

Answer: Done.

  1. Line 384: now is „32 specimens”, should be „621 specimens”

Answer: Done.

Reviewer 2 Report

Dear  authors,

thank you very much for an improved version of the ms.

I have minor comments. 

1) This paper is the fifth dedicated exclusively to the taxon Collembola in relation with this study (Redundant information for readers. Please, remove this part.)

Answer: It does not seem redundant information because it has not been said before anywhere in the text.

In different context, I think that it is redundant for section „Simple Summary“

2) Previous studies were focused on ecology [3–4], or 35 faunal aspects [5–7] (I have no doubt, that authors are well informed about fauna from MSS habitat, please, can you add several recent references relevant to this statement, e.g. Rendoš, M., Raschmanová, N., Kováč, Ľ., Miklisová, D., Mock, A., Ľuptáčik, P., 2016: Organic carbon content and temperature as substantial factors affecting diversity and vertical distribution of Collembola on forested scree slopes. European Journal of Soil Biology, 75: 180–187; Jureková, N., Raschmanová, N., Miklisová, D., Kováč, Ľ., 2021: Mesofauna at the soil-scree interface in a deep karst environment. Diversity, 13(6), 242.

Answer: We must remember that Jurekova's paper refers to a karstic environment, very different from that of the Sierra de Guadarrama, which is Granitic. No reference has been made to other papers referring to karstic MSS because this biotope is granitic. In a karstic environment, there are caves and the scree-slope can be an intermediate place between the caves and the outside environment. The springtails that live in caves are not really from caves but from the infractions of the karstic terrain, which by definition is very broken. There are no caves in terrains such as the MSS de Guadarrama, and the only place far from the surface is the bottom of about two meters of the scree-slope. The fauna found in it is mainly its own fauna, as has been shown in our previous works.

In relation to this paper (Rendos), we have not included these factors (coal and temperature) in our studies nor have we made any relation to them.

We have included an additional paragraph in the General Discussion to make it clear what the context of our study is. We appreciate the remark made.

OK- regarding additional paragraph in the General Discussion, but first of all, this is needed to specify here, in „Introduction“. Perhaps you can rephrase to be more accurate.

3) The first initial period...ok, please to give information about second period, only reference about sampling desiguan, this is insufficent for reader.

Answer. As has been said in methodology, in this second period the same traps have been used as in the first period and with the same sampling system. In addition, there is expanded information in the earlier works cited in this one.

I still have a poblem that authors often refer to previous published studies in this ms. There is important to note more details in section „Methods“, really useful for reader.

Author Response

Reviewer 2 said:

Dear  authors,

thank you very much for an improved version of the ms. I have minor comments.

1) This paper is the fifth dedicated exclusively to the taxon Collembola in relation with this study (Redundant information for readers. Please, remove this part.)

Answer: It does not seem redundant information because it has not been said before anywhere in the text.

In different context, I think that it is redundant for section „Simple Summary“

Answer: Done

2) Previous studies were focused on ecology [3–4], or 35 faunal aspects [5–7] (I have no doubt, that authors are well informed about fauna from MSS habitat, please, can you add several recent references relevant to this statement, e.g. Rendoš, M., Raschmanová, N., Kováč, Ľ., Miklisová, D., Mock, A., Ľuptáčik, P., 2016: Organic carbon content and temperature as substantial factors affecting diversity and vertical distribution of Collembola on forested scree slopes. European Journal of Soil Biology, 75: 180–187; Jureková, N., Raschmanová, N., Miklisová, D., Kováč, Ľ., 2021: Mesofauna at the soil-scree interface in a deep karst environment. Diversity, 13(6), 242.

Answer: We must remember that Jurekova's paper refers to a karstic environment, very different from that of the Sierra de Guadarrama, which is Granitic. No reference has been made to other papers referring to karstic MSS because this biotope is granitic. In a karstic environment, there are caves and the scree-slope can be an intermediate place between the caves and the outside environment. The springtails that live in caves are not really from caves but from the infractions of the karstic terrain, which by definition is very broken. There are no caves in terrains such as the MSS de Guadarrama, and the only place far from the surface is the bottom of about two meters of the scree-slope. The fauna found in it is mainly its own fauna, as has been shown in our previous works.

In relation to this paper (Rendos), we have not included these factors (coal and temperature) in our studies nor have we made any relation to them.

Answer: We have included an additional paragraph in the General Discussion to make it clear what the context of our study is. We appreciate the remark made.

OK- regarding additional paragraph in the General Discussion, but first of all, this is needed to specify here, in „Introduction“. Perhaps you can rephrase to be more accurate.

Answer: Done

3) The first initial period...ok, please to give information about second period, only reference about sampling desiguan, this is insufficent for reader.

Answer. As has been said in methodology, in this second period the same traps have been used as in the first period and with the same sampling system. In addition, there is expanded information in the earlier works cited in this one.

I still have a poblem that authors often refer to previous published studies in this ms. There is important to note more details in section „Methods“, really useful for reader.

Answer: Done. A new Figure (1), and a new sentence have been added.